# The ActiveText@T2D text messaging behavioural intervention to increase physical activity in adults with type 2 diabetes: A prospective single-arm feasibility trial

Holly Blake[1,2]☯*, Mohammed Jameen Alsahli[1,3]☯, Wendy J. Chaplin[1,2], Stathis Th. Konstantinidis[1]

**1** School of Health Sciences, University of Nottingham, Nottingham, United Kingdom, **2** NIHR Nottingham Biomedical Research Centre, Nottingham, United Kingdom, **3** Division of Health Informatics, Saudi Electronic University, Riyadh, Saudi Arabia

☯ These authors contributed equally to this work.
* holly.blake@nottingham.ac.uk

## Abstract

Physical activity is a core aspect of type 2 diabetes (T2DM) self-management, but most Saudi adults do not meet physical activity recommendations and there are no culturally tailored interventions to promote physical activity in Saudi adults with T2DM. This study is a prospective single-centre, single-arm feasibility study of a mobile SMS text messaging intervention with a nested qualitative study. The aim was to explore the feasibility and acceptability of ActiveText@T2D, a 6-week theory-based mobile text messaging intervention to promote physical activity in people with T2DM in Saudi Arabia. Intervention development was informed by the Behaviour Change Wheel (BCW) framework and COM-B model. ActiveText@T2D consisted of 2 one-way SMS text messages per week, for 6 weeks. All participants were offered the intervention and assessed at baseline (Time 0: T0) and 3-month follow-up (Time 1: T1). Data collection included feasibility outcomes (recruitment and retention), clinical outcomes (body mass index and glycaemic control from clinic records at T0), and self-reported outcomes (self-efficacy, physical activity, and barriers to exercise at T0, T1). Qualitative interview data (n = 19) were collected at T1 with 11 patients (7 male, 4 female, mean age 54.5 years) and 8 female nurses (mean age 31.8 years). Quantitative data were analysed descriptively, qualitative data were analysed thematically. Of 98 participants approached, 62 were eligible, and 52 consented (84% participation rate; 23 women, 29 men; mean age 54.82 years), 44 (85%) completed baseline measures and received the intervention. Thirty-nine participants completed follow-up measures (75% retention to T1). All outcome measures were sensitive to change: The Arabic version of the CDC Barriers to Being Active Quiz (BBAQ), The Arabic version of Exercise Self-Efficacy scale (ESE-A), The Arabic International Physical Activity Questionnaire (A-IPAQ). Patients and healthcare professionals perceived the

**Data availability statement:** The dataset used in this study is available from the University of Nottingham Research Data Repository (http://doi.org/10.17639/nott.7513).

**Funding:** The study was funded by Ministry of Higher Education, Saudi Arabia (awarded to MJA, Ref: 1070420532). The funder had no role in the study design, data collection and analysis, decision to publish, or preparation of the manuscript.

**Competing interests:** The authors have declared that no competing interests exist.

intervention to be broadly acceptable. Qualitative findings identified three overarching themes: "use of text messaging as a health intervention", "engagement with physical activity" and "instilling knowledge about physical activity and diabetes control". This study demonstrates the feasibility and acceptability of ActiveText@T2D, a theory-based culturally tailored SMS text messaging intervention, to Saudi adult patients with T2DM and healthcare professionals involved in their care. The next step would be a full-scale definitive randomised controlled trial to assess the effectiveness and cost-effectiveness of ActiveText@T2D.

**Protocol registration:** Protocols.io, DOI: dx.doi.org/10.17504/protocols.io.261ger217l47/v1 (registered on 08.01.2025).

## Author's summary

The population prevalence of poorly controlled type 2 diabetes mellitus (T2DM) in Saudi Arabia is high. Physical activity is an important part of diabetes management, but many people with T2DM do not meet the recommended physical activity recommendations. We developed ActiveText@T2D, a 6-week theory-based culturally tailored mobile text messaging intervention aimed at promoting physical activity in people with T2DM in Saudi Arabia. In a trial with 52 participants, our study found that ActiveText@T2D is feasible and acceptable to patients with T2DM, and healthcare professionals involved in their care. The intervention may increase physical activity participation and self-efficacy for exercise, and reduce barriers to physical activity, although findings need to be tested in a randomised clinical trial. Our study generated important insights about the value of text messaging for health promotion as an adjunct to diabetes clinical care, and the need for communication with patients about physical activity, which is perceived to be lacking in clinical consultations.

## Background

Diabetes is a serious public health concern, associated with significant social, financial and health system burden, worldwide [1]. Considered a 'growing epidemic' [2], diabetes is associated with high morbidity, poor health-related quality of life [3,4] and is considered a major global cause of premature mortality [3,5]. The economic burden of diabetes is large [6,7], accounting for 1.8% (1.7-1.9) of global GDP in 2015, anticipated to rise to 2.2% (2.1–2.2) by 2030 [7]. This burden is associated with direct medical costs of diabetes and its complications (e.g., blindness, kidney failure, heart attacks, stroke and lower limb amputation [3]), and indirect costs such as sickness absenteeism, productivity loss, reduction in labour force participation and mortality [7].

In 2019, the global prevalence of diabetes mellitus (DM) was estimated to be 9.3% (463 million people), predicted to rise to 10.9% (700 million people) by 2045 [8]. The prevalence of DM is particularly high in Arab Gulf Cooperation Council countries [9]. In 2017, the World Health Organization ranked Saudi Arabia as having the second highest rate of diabetes in the Middle East and 7th highest in the world, with around 7 million people living with diabetes and more than 3 million with pre-diabetes [9,10]. Although there are geographical variations, the overall pooled prevalence of type 2 diabetes mellitus (T2DM) in Saudi Arabia has recently been estimated at 16.4% [11]. Myriad reasons for this high prevalence include an ageing population, rising obesity, unhealthy diet, physical inactivity, and genetic disposition [12]. The rising population of people with diabetes in Saudi Arabia is further at risk for diabetes complications and mortality, since studies show that 75% has uncontrolled diabetes (i.e., HbA1c ≥ 7%; [13]).

Studies of Saudi T2DM patients have revealed low health literacy [14–16] which contributes to poorly managed diabetes. This highlights an urgent need for intervention to support self-management behaviours in Saudi individuals with T2DM.

Physical activity is important for the health of all adults [17] and alongside dietary changes is typically one of the first self-management strategies advised for people with T2DM [18]. However, physical inactivity is common in the Saudi population, and most Saudi adults are not active enough to meet the recommended guidelines for physical activity [19,20]. Studies of physical activity in Saudi diabetes populations are limited, but indicate low proportions are meeting physical activity recommendations of 150 minutes of moderate activity per week (e.g., 26.3%: [21]; 30%: [13]).

Barriers to physical activity in Saudi Arabia include lack of time, lack of confidence, environmental factors (e.g., high density traffic and poor air quality, high temperature outdoors, lack of suitable facilities), gender (i.e., being female), social factors (e.g., lack of support, family/caregiving responsibilities), individual factors (e.g., self-efficacy, enjoyment, motivation, willpower, fear of injury, perceived safety), cultural barriers to exercise (e.g., traditions) and for those with diabetes, longer disease duration [21–28]. Interventions are needed that promote physical activity in Saudi adults, while addressing culturally relevant barriers to engagement.

Technology has been used to promote health behaviour change for many years [29]. One approach is mobile health (mHealth), defined by the World Health Organization's (WHO) Global Observatory for eHealth as "medical and public health practice supported by mobile devices, such as mobile phones, patient monitoring devices, personal digital assistants (PDAs), and other wireless devices" [30]. Mobile phone ownership is ubiquitous; 91% of the Saudi population aged 12–65 years (73.28% of the total population) owns a mobile phone. While mHealth research is advancing rapidly due to its flexibility and scalability, most advanced mHealth solutions (e.g., apps, and wearable devices) have been implemented in developed countries. Health interventions need to be context specific [31] and therefore the nature of existing mHealth interventions, and findings from such studies, may not directly translate to developing countries in which lived experiences, values, beliefs, and cultures are likely to be different. Albeit with a fast-growing economy, Saudi Arabia is considered a developing country.

Research shows that mHealth interventions may be effective for increasing physical activity in adult populations [32–34] including those with existing health conditions or at risk for ill-health [35]. Specifically, mobile text messaging interventions are effective in increasing physical activity in people with and without chronic health conditions [36]. However, there are very few studies of text messaging to promote physical activity in T2DM [37]. Although text messaging physical activity interventions showed promise in an early review [38], there remains inconclusive evidence of their effectiveness, and there are no prior studies with Saudi populations that focus solely on physical activity promotion in T2DM [37]. In Saudi Arabia, mobile phones are a socially popular form of communication, but mobile-based health interventions (including text messaging) remain highly novel as a health promotion approach. However, the feasibility and acceptability of this approach needs to be examined before a definitive trial can be conducted.

## Aims and objectives

The aim of this study was to explore the feasibility and acceptability of a theory-based mobile text messaging intervention to promote physical activity in people with T2DM in Saudi Arabia. The primary objectives were (i) to determine feasibility based on quantitative rates of recruitment to the study, retention rates, the success of transmitting text messages, and completion of outcome measures, and (ii) to determine acceptability and participant engagement with the intervention through qualitative exploration of patients' and healthcare professionals' (HCP) perceptions towards the intervention. A secondary objective was to assess any changes after the intervention, compared to baseline, in quantitative measures of self-efficacy, physical activity, glycated haemoglobin (HbA1c) levels, and body mass index (BMI).

## Intervention development

The ActiveText@T2D intervention was designed to increase physical activity participation among Saudi adults with T2DM, to contribute towards T2DM management. Content development was informed by the Behaviour Change Wheel (BCW) framework and COM-B model [39,40]. The COM-B model suggests that people need capability (C), opportunity (O) and motivation (M) to perform a new behaviour (B). *Capability* is an individual's psychological and physical ability to participate in an activity, *opportunity* refers to external factors that make a behaviour possible, and *motivation* refers to the conscious and unconscious cognitive processes that direct and inspire behaviour [39].

APEASE criteria were applied to determine whether it was possible for Behaviour Change Techniques (BCTs) to be implemented within the targeted behaviour (physical activity), target population (patients with T2DM), context (Saudi Arabia) and the study setting (community). A BCT is defined as a replicable component of an intervention designed to alter or redirect causal processes that regulate behaviour (i.e., a technique is proposed to be a potentially "active ingredient") [41]. Using the behaviour change techniques taxonomy, two intervention functions (education and persuasion) were selected to design the text messaging content which was mapped to relevant BCTs (Table 1).

A set of 13 text messages was developed in English by the researcher (a Saudi male, with expertise in health informatics) and reviewed for content validity by an expert panel (n = 6) with expertise in health psychology, nursing, endocrinology/diabetes care, mHealth and health informatics. Two members were fluent in English, four members were fluent in English/Arabic and familiar with Saudi culture. Messages were culturally tailored for the target population, but were not personalised, titrated, or adapted during the intervention period. A toolkit was adapted from the Model Systems Knowledge Translation Center resources on Diversity, Equity, Inclusion, and Accessibility for disability and rehabilitation

**Table 1. Mapping intervention functions and BCTs.**

| Intervention functions | BCTs | BCTs definition |
|---|---|---|
| Education | Provide information about behaviour | General information about behavioural risk (e.g., susceptibility to poor health outcomes or mortality risk in relation to the behaviour. |
| | Provide information on consequences | Information about the benefits and costs of action or inaction, focusing on what would happen if the person does or does not perform the behaviour. |
| Persuasion | Set graded tasks | Set easy tasks and increase difficulty until target behaviour is performed. |
| | Provide instruction | Telling the person how to perform a behaviour and/or preparatory behaviours. |
| | Prompt specific goal setting | Involves detailed planning of what the person will do, including a definition of the behaviour, specifying frequency, intensity, or duration and specification of at least one context, that is, where, when, how, or with whom. |
| | Provide general encouragement | Praising or rewarding the person for effort or performance without this being contingent on specified behaviours or standards of performance. |

BCTs: Behaviour Change Techniques.

researchers [42]. The panel was required to assess the developed text messages against the following toolkit criteria: the content relevance (appropriateness of each message to the theoretical constructs), readability (appropriate reading level), and tone (matching of the messages to the response level) [42]. The final message set (S1 Table) was then reviewed by a patient and public involvement and engagement (PPIE) group consisting of three Saudi patients with T2DM to ensure message content was understandable, relevant, and culturally appropriate. Messages were developed in English and translated to Arabic (for patient review and delivery) by a bilingual translator, using forward-back translation processes described by Son [43]. Text messages were limited to 160 characters to prevent them from being automatically converted to Multimedia Messaging Service (MMS). The development processes resulted in minor revisions to message ordering and sentence structure.

## Materials and methods

### Study design

This study is a single-centre, single-arm feasibility study of a mobile text messaging intervention with a nested qualitative study. The study protocol was informed by the Standard Protocol Items: Recommendations for Interventional Trials (SPIRIT) statement [44]. Reporting of trial results is guided by the Consolidated Standards of Reporting Trials (CONSORT) extension to randomised pilot and feasibility trials [45] (S2 Table), and the Template for Intervention Description and Replication (TIDieR) checklist (S3 Table) [46]. All participants were offered the 6-week intervention and assessed at baseline (Time 0: T0) and 3-month follow-up (Time 1: T1). The study protocol was registered on Protocols.io on 08 January 2025 (dx.doi.org/10.17504/protocols.io.261ger217l47/v1).

### Study setting and participants

As this is a feasibility study, power calculation for sample size determination is not required [47]. Participants with T2DM were recruited from the diabetes and endocrinology outpatient clinic of a large public hospital in Saudi Arabia. Usual care for patients with T2DM at the participating site includes three clinic visits per year, consisting of medical consultations and diabetes education delivered by nurses, clinical educators, and dieticians. Data collection took place in the hospital setting, while the mHealth intervention took place in participants' own homes. Participant flow through the study is shown in Fig 1.

### Participant eligibility

Eligible participants had a clinical diagnosis of T2DM, according to the American Diabetes Association (ADA) criteria [48] and were attending a hospital diabetes and endocrinology clinic.

• Inclusion criteria: Participants were included if they were 18 years of age or over, had access to a mobile phone with text messaging capability, and could read and understand Arabic.

• Exclusion Criteria: Participants were excluded if they were unable to give informed consent, were on insulin adjustment treatment, were visually impaired, had diabetes complications or were medically unable to perform moderate physical activity as assessed by a physician.

### Ethics

The protocol received a favourable ethical opinion from the Faculty of Medicine and Health Sciences Research Ethics Committee at the University of Nottingham on 25 October 2019 (Ref: 414–1910) and King Fahd Medical City (KFMC) in Saudi Arabia on 03 November 2019 (Ref: H-01-R-012).

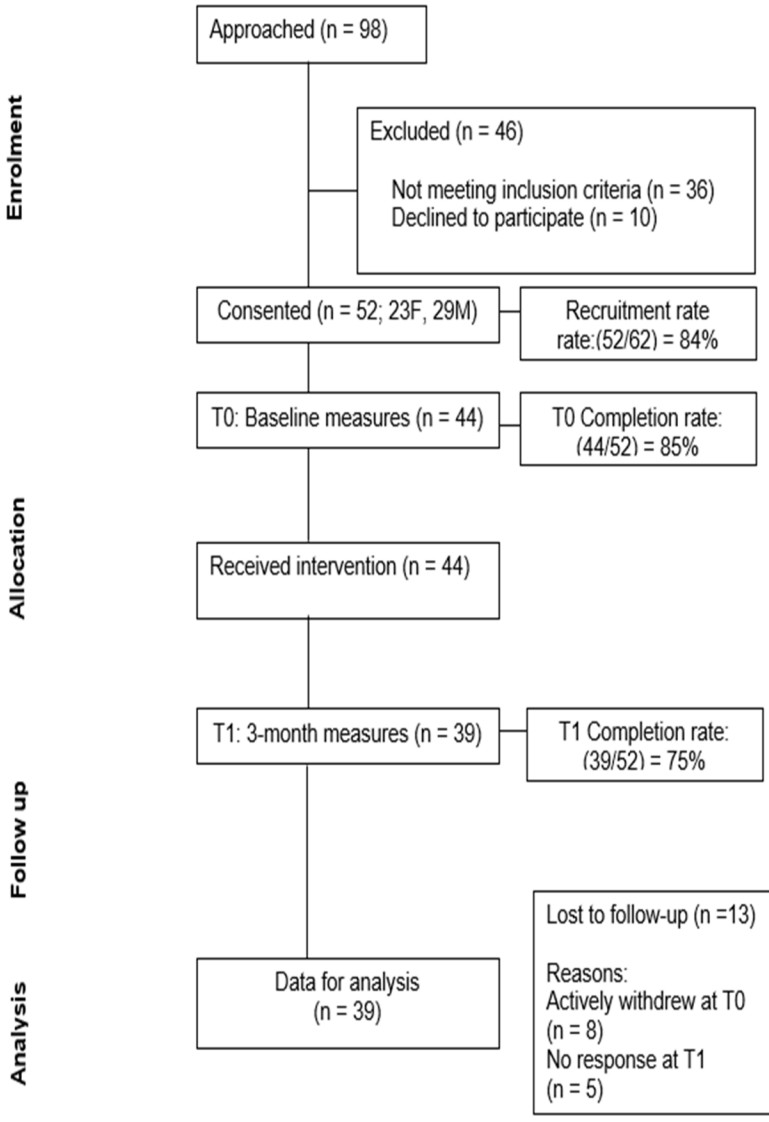

**Fig 1. CONSORT flow diagram.**

## Recruitment

Participants were recruited through direct approach to consecutive clinic attendees. This approach was chosen since our PPIE work indicated that recruiting through face-to-face interaction is deemed to be a more respectable way to communicate in Saudi culture, compared to written invitation. Eligibility was assessed by the clinical care team over a two-week period between 22 December 2019 and 5 January 2020, to screen out ineligible participants. Prospective patients were approached in one of two ways, either by one of two nurses while in the waiting room prior to clinic appointment, or by their healthcare provider at the end of their routine appointment. All staff taking informed consent received the same guidance from the study researcher on recruitment, consent, and trial procedures. Potential participants were provided with (a) a study information sheet (in the Arabic language) describing the study purpose, length, perceived benefits and contact details for the researcher, and (b) a consent form. Written informed consent was taken by the researcher (MJA) or

a nurse. Participants had the option to provide consent immediately onsite or take the study information home to discuss with family or friends. Those who did not consent on the day of approach were followed up after 24 hours by WhatsApp message sent from the research team (for those who provided consent to share their mobile phone number for follow-up). All eligible participants were informed that their participation was voluntary and that they may withdraw from the study at any time without prejudice to their care. Participants who declined primarily did so due to personal concerns about health issues, functional illiteracy, concerns about data confidentiality, lack of interest, family refusal or unwillingness to engage in intervention or follow-up procedures. Treatment and care from the hospital team continued as usual for all participants during the intervention. Retention strategies included: (1) participants individually thanked for their time, (2) option of paper or online surveys (via web link) at T0, at participant preference, (3) Two WhatsApp text message reminders to complete T1 questionnaires, (4) an opt-in shopping voucher the equivalent of a coffee (Saudi Riyal: SR 15), for T1 completers. As the data collection period did not coincide with the patient's clinic appointments at T1, data were collected using an online survey via a link sent on WhatsApp. There were no changes to the methods, eligibility criteria or measures after the feasibility trial commenced.

### Intervention delivery

The text messages were programmed for one-way automated transmission using a commercial SMS provider (https://www.mobily.ws/en/), the primary authorised provider in Saudi Arabia at the time of the study. The system did not enable two-way messaging. The system was composed of an SMS data collection tool which uses an in-built SMS gateway to send one-way structured messages that are then stored in a local database and utilise an enabled analytics tool that presents data visualisation to track the trial progress of sent messages. In total, 13 messages were sent to each participant, at a rate of two text messages per week, over a six-week period. This *duration* of messaging is based on the preference expressed by our PPIE group, coupled with prior research suggesting that an average of 18–60 days is required for a new behaviour to become a habit [49]. Regarding message *frequency*, there is no consensus on the most appropriate number of messages to send, and this varies greatly in prior research from once weekly to 5 or more times per day [50]. However, prior research by the lead author (HB) has found that two messages sent twice per week is acceptable to patients with chronic conditions [49], and this approach was agreed upon by our ActiveTextT2D PPIE group. In terms of message *timing*, Our PPIE group preferred that messages were sent at various times (i.e., morning: 9:30 am; evening: 7:30 pm) on Mondays, Wednesdays, and Thursdays. The schedule for text message delivery considered the context of life in Saudi Arabia (i.e., people pray five times throughout the day). The PPIE group also indicated that it is preferably to receive the text messages during late morning or early evening hours to avoid exposure to environmental factors (i.e., hot weather) that could demotivate individuals to participate in exercise, such as walking. Prior studies using text messaging to promote behaviour change in other medical conditions have advocated early morning and early evening hours [51]. The intervention content and delivery were not modified during the study. Participants could report any technical issues, or requests to discontinue the intervention, directly to the researcher using telephone, WhatsApp, or email, at their preference.

### Feasibility outcomes and measures

Feasibility outcomes (Table 2) were recorded, including recruitment, retention, intervention fidelity, safety, and acceptability. Demographic data were collected by self-report questionnaire at T0. This included age, gender (male/female), diagnosis, medical history, marital status (single/married/separated/divorced/widowed), ethnicity (Saudi, Asian, Caucasian, Latino). Health data were accessed from the clinical notes at T0, including body mass index (BMI) and an indicator of glycaemic control (HbA1c). Participants completed questionnaire measures at T0 and T1 to assess self-reported physical activity, exercise self-efficacy, and barriers to physical activity. Measures included:

**Table 2. Feasibility and acceptability measures.**

| Data collection | Measurement Methods | Details | Purpose | T0 | T1 |
|---|---|---|---|---|---|
| **i. Recruitment** | Study records | The number of participants approached and consented, the characteristics of those recruited any barriers to recruitment, and the timescale to recruit. | To assess whether it would be possible to recruit to a definitive trial. | ✓ | |
| **ii. Retention** | Study records | The number of participants who complete outcome measures at T0 and T1. | To assess whether it would be possible to retain participants in a definitive trial. | ✓ | ✓ |
| **iii. Intervention fidelity** | Text Message System Provider | Fidelity (number of messages sent and delivered). | To assess fidelity of the intervention. | | ✓ |
| | Study records | Technical issues, self-reported by participants via telephone, WhatsApp, or email. | To assess degree of engagement with the intervention and how this resource may be used in a definitive trial. | ✓ | ✓ |
| | Interviews (participants) | Engagement with and perceptions of the intervention. | To explore perceptions of the intervention, engagement with the intervention, and any barriers/challenges and reported impacts on behaviour change. | | ✓ |
| | Interviews (nurses) | Perceptions of the intervention and alignment with clinical care. | To explore perceptions of the intervention, any observed barriers/challenges or observed impacts on behaviour change, and alignment with clinical care. | | ✓ |
| **iv. Safety** | Study records | Adverse event monitoring during the intervention period (e.g., directly related to any increased physical activity as recommended in the intervention). | To assess the feasibility of safety data capture for a definitive trial. | ✓ | ✓ |
| **v. Acceptability** | Interviews (participants and nurses) | Qualitative interviews will explore whether participants and nurses find the intervention and trial design acceptable. | To assess the acceptability of using this intervention and design in a definitive trial. | | ✓ |

Abbreviations: T0 – Timepoint 0 (baseline); T1 – Timepoint 1 (3 months).

(a) Arabic International Physical Activity Questionnaire (A-IPAQ): The IPAQ [52] is a measure of physical activity, with established validity [52–55]. The IPAQ has been culturally adapted for Arabic-speaking countries (in the Middle East and North Africa region). The 7-item Arabic version of the long-form International Physical Activity Questionnaire (A-IPAQ) [56] was used in this study. It has demonstrated acceptable validity and reliability for the assessment of physical activity among Arabic adults [56]. The A-IPAQ assesses physical activity in four domains: occupational (related to work), domestic (house chores), transportation (walking, public transportation) and leisure time (recreational activities) and differentiates between usual sitting time on a weekday and a weekend day. Overall scores are calculated using responses to all questions, with sub-scores calculated for (1) Walking, (2) Moderate-intensity activity, (3) Vigorous-intensity activity, (4) Sedentary activity.

(b) The Arabic version of Exercise Self-Efficacy scale (ESE-A): The ESE-A is an 18-item measure for determining the level of exercise self-efficacy. It has established reliability and validity (Cronbach's $\alpha = 0.89$)[57]. The scale items measure exercise self-efficacy in various situations, with responses ranging from "0" (can't do exercise) to "100" (certainly can). The total scores for exercise self-efficacy are calculated as the sum of all scale items, divided by the total number of items. A higher score indicates a higher level of confidence in performing the exercise.

(c) The Arabic version of the CDC Barriers to Being Active Quiz (BBAQ): Barriers to physical activity were measured using the Barriers to Being Active Quiz (BBAQ) from the Centers for Disease Control and Prevention [58] Diabetes Road to Health Toolkit. The instrument has a total of 21 items across 7 domains, including five internal barriers (lack of time, lack of energy, lack of willpower, fear of injury, and lack of skill) and two external barriers (social influence and lack of resources). Each domain contains three items, with a total score range of 0–63. Respondents rate the degree

of activity interference on a 4-point scale ranging from 0 (very unlikely), 1 (somewhat unlikely), 2 (somewhat likely), to 3 (very likely). A score of 5 or above in any category indicates that the domain is a critical barrier for the individual. The instrument has been used previously with T2DM patients and with Saudi adults (e.g., [59,60]).

### Quantitative data analysis

Quantitative data were analysed using the Statistical Package for the Social Sciences Version 25.0 (Armonk, NY: IBM Corp.). Feasibility outcomes were reported using frequencies (n%), means, standard deviations, and ranges. Pre-specified criteria to move forwards to a definitive trial include: > 50 of eligible patients consented, > 70% retention to T1, > 90% of messages successfully sent. No interim analyses were planned. Stopping criteria included any serious adverse events. To explore whether the A-IPAQ, ESE-A, and BBAQ outcome measures were sensitive to change, pre- and post-scores were compared for each research measure, using Wilcoxon-signed rank test. A significance threshold ($p < 0.05$) was determined for all statistical tests. All outcome measures were included in analyses, and missing data were reported. The effect size, measured by rank-biserial correlation and calculated in R using the 'rstatix' package. Effect size (r) was interpreted as small effect = 0.10, moderate effect = 0.30, and large effect >0.50 effect [61].

### Inclusivity in global research

Additional information regarding the ethical, cultural, and scientific considerations specific to inclusivity in global research is included in the Supporting Information (S1 Checklist).

## Results

### Feasibility outcomes

Recruitment: Of the 98 participants approached, 36 did not meet the eligibility criteria. Of the 62 eligible participants, 10 declined to participate. Fifty-two patients with T2DM consented to take part (23 women, 29 men; 84% uptake from those eligible). Recruitment took place between 22 December 2019 and 5 January 2020, at a rate of approximately 4 patients per day. Baseline measures were completed during the 2-week recruitment period. Age ranged from 29 to 71 years; mean age of participants was 54.82 years (SD = 11.4). Most participants were of Saudi Arabian nationality (98%) with one from Jordan (2%). At T0, 63.5% of participants had co-morbid conditions (38.5% hypertension, 25% other). BMI ranged from 22.6 to 44.1; mean BMI was 30.7 (SD = 5.1). Regarding glycaemic control, HbA1c ranged from 5.6 to 12.1 (mean 7.6, SD = 1.3). Most participants were married (83%), and 92% were from moderate-to-high-income households. Almost half (44.2%) were educated beyond high school (having a college or university education). Barriers to recruitment were observed in the clinic setting, which included limited time to explain the study and obtain informed consent, concerns about privacy in public waiting rooms, and family members influencing participants' decisions to participate. The data are shown in the participant's demographic characteristics Table 3.

(i) Retention: 52 participants were recruited, and their demographics were recorded. However, 8 withdrew before completing the T0 measures, for the same reasons described above as recruitment barriers. All participants who completed T0 measures received the intervention (n = 44). At T1, 39 participants completed the follow-up questionnaire measures, and five participants dropped out (75% retention to T1). No participants reported issues with study burden (overall, relating to the measures and intervention) although 3 reported that the questionnaires were lengthy. All T1 measures were completed within 3 weeks of the study end.

(ii) Intervention fidelity: The intervention was delivered as planned. A total of 697 text messages were sent to participants over a period of 6 weeks. Of the 697 text messages, 668 (96%) were recorded as being successfully sent, 29 text messages were not delivered. The automated system did not report their reasons for being missing, but participant self-reports suggested it may relate to lack of mobile phone signal.

(iii) Safety: No adverse events were reported.

**Table 3. Participant characteristics at baseline (n = 52).**

| ID | Sex | Age | Education level | Marital Status | Income Level | Other health conditions | BMI | Glycated haemoglobin HbA1c |
|---|---|---|---|---|---|---|---|---|
| 5 | Male | 55 | Undergraduate | Married | Moderate | Hearing problems, low mood | 31.06 | 10.8 |
| 7 | Male | 38 | Postgraduate | Married | High | Dyslipidemia | 38.03 | 5.7 |
| 9 | Female | 41 | High school | Married | Moderate | – | 36.65 | 7.8 |
| 10 | Male | 63 | Undergraduate | Married | High | – | 32.62 | 8.2 |
| 11 | Female | 49 | High school | Widowed | Moderate | Bronchial asthma | 29.92 | 7.2 |
| 12 | Male | 44 | High school | Married | Moderate | Hypertension | 29.58 | 8.6 |
| 13 | Male | 58 | Postgraduate | Married | High | – | 32.13 | 7.6 |
| 14 | Male | 31 | Undergraduate | Single | Moderate | – | 25.75 | 5.6 |
| 15§ | Male | 66 | High school | Married | Moderate | Hypertension | 25.00 | 7.1 |
| 16# | Male | 65 | Middle school | Married | Moderate | – | 26.23 | 6.2 |
| 17 | Male | 37 | Undergraduate | Married | Moderate | – | 26.95 | 8.4 |
| 18 | Male | 29 | Undergraduate | Single | Low | – | 36.33 | 6.4 |
| 19# | Female | 43 | High school | Married | Moderate | – | 38.08 | 6.9 |
| 20 | Female | – | Undergraduate | Married | Low | – | – | – |
| 21 | Female | 44 | High school | Married | Moderate | Thyroid problem | 33.81 | 6.2 |
| 22 | Male | 36 | High school | Married | Moderate | Hodkin's lymphoma | 31.17 | 6.7 |
| 23 | Female | 69 | Postgraduate | Divorced | High | Hypertension | 25.65 | 6.8 |
| 24# | Male | 65 | Undergraduate | Married | Moderate | Hypertension, Bronchial asthma | 31.12 | 7.8 |
| 26 | Female | 36 | Postgraduate | Married | High | | 34.09 | 6.1 |
| 27 | Female | 64 | Undergraduate | Married | Moderate | Hypertension, Osteoporosis | 37.72 | 8.8 |
| 28# | Female | 66 | Middle school | Widowed | Moderate | Dyslipidemia | 26.78 | 7.6 |
| 29 | Male | 64 | Undergraduate | . | High | – | – | – |
| 30 | Male | – | Undergraduate | . | High | – | – | – |
| 31 | Male | 65 | Undergraduate | Married | High | – | 24.65 | 7.0 |
| 32 | Male | 58 | Postgraduate | Married | Moderate | – | 26.08 | 8.1 |
| 33 | Female | 58 | Postgraduate | Married | High | Hypertension, Stroke, Deep vein thrombosis, systemic lupus erythematosus (SLE) and seizure | 33.78 | 12.1 |
| 34 | Male | 71 | No education | Married | Moderate | Hypertension | 24.02 | 8.4 |
| 35# | Female | 60 | No education | Married | Low | – | 38.15 | 8.7 |
| 36# | Male | 65 | Middle school | Married | Moderate | Hypertension | 28.72 | 9.9 |
| 37§ | Male | 43 | High school | Married | Moderate | Liver disease | 28.86 | 9.1 |
| 38 | Male | 61 | Undergraduate | Married | High | Hypertension | 33.27 | 9.0 |
| 39# | Female | 65 | No education | Married | Moderate | Dyslipidemia | 28.64 | 6.7 |
| 40 | Female | 39 | High school | Married | Moderate | Thyroid problem | 35.57 | 5.9 |
| 41 | Female | 44 | Middle school | Married | Moderate | – | 37.04 | 8.8 |
| 42 | Male | 59 | Undergraduate | Married | Moderate | Dyslipidaemia, Hypertension | 27.78 | 6.6 |
| 43 | Male | 53 | High school | Married | Moderate | Hypertension | 28.60 | – |
| 44 | Male | 53 | High school | Married | Moderate | – | 23.46 | 7.4 |
| 46§ | Female | 48 | High school | Married | Moderate | Hypertension | 24.68 | 7.2 |
| 47 | Male | 71 | Middle school | Married | Moderate | Hypertension | 30.98 | 7.8 |
| 48 | Male | 67 | Postgraduate | Married | High | Hypertension | 24.38 | 7.4 |
| 49§ | Female | 62 | Middle school | Married | Moderate | – | 22.62 | 7.4 |
| 50 | Female | – | – | – | – | – | | |
| 51 | Male | 52 | Middle school | Married | Moderate | – | 22.60 | 6.7 |
| 52 | Female | 52 | Postgraduate | Married | Moderate | Overweight, knee pain | 38.29 | 6.9 |

*(Continued)*

**Table 3.** (Continued)

| ID | Sex | Age | Education level | Marital Status | Income Level | Other health conditions | BMI | Glycated haemoglobin HbA1c |
|---|---|---|---|---|---|---|---|---|
| 53 | Female | 59 | Middle school | Married | Moderate | Overweight, ambulatory | 28.86 | 8.3 |
| 54 | Male | 67 | Postgraduate | Married | High | Hypertension | 36.96 | 6.9 |
| 55 | Female | 61 | No education | Married | Moderate | Hypertension, meningioma | 32.12 | 8.0 |
| 56# | Female | 60 | No education | Married | Moderate | Hypertension | 29.86 | 6.9 |
| 57 | Female | 59 | High school | Married | Moderate | Hypertension | 44.05 | 8.7 |
| 58§ | Male | 65 | No education | Married | Moderate | Anxiety | 28.73 | 6.8 |
| 59 | Female | 61 | High school | Married | Moderate | Hypertension, endocrine problems | 30.41 | – |
| 60 | Male | 45 | Undergraduate | Widowed | Moderate | Hypertension | – | 7.3 |

#After recruitment, the participant withdrew, before completing T0 questionnaires (n = 8). §Participant completed T0, but not T1 measures (n = 5). BMI: Body mass index. Missing data age (n = 3), education (n = 1), marital status (n = 3), income level (n = 1), BMI (n = 5), and HbA1c (n = 6).

## Questionnaire outcomes

The purpose of a feasibility study is not to assess intervention effectiveness. However, pre-post comparisons were made using non-parametric Wilcoxon tests for paired samples to explore whether the A-IPAQ, ESE-A (Table 4) and BBAQ (Table 5) outcome measures were sensitive to change. Paired data was used to compare participant data at baseline and follow-up data and used complete cases only (n = 39/44; 5 dropouts from T0 to T1). Missing data for A-IPAQ: MET-vigorous physical activity (n = 2), MET–moderate physical activity (n = 5), and MET-walking (n = 5). BBAQ scores had missing pairs in the following domains: social influence (n = 1), lack of energy (n = 3), lack of willpower (n = 2), fear of injury (n = 2), lack of skill (n = 4), lack of resources (n = 2). There were no missing data for lack of time or the three environmental factors. For A-IPAQ, median MET (Metabolic Equivalent Task: the ratio of the rate of energy expended during an activity to the rate of energy expended at rest), MET Moderate Activity and MET Vigorous Activity increased from T0 to T1 (Table 4). The increase was statistically significant for MET Moderate Activity and MET Vigorous Activity, with large effect sizes [r = 0.86 (95%CI 0.60 to 1.12) and r = 0.79 (95%CI 0.54 to 1.04) respectively]. There was no statistically significant difference in the median MET Walking at the two time points or the difference in the time spent sitting during weekdays; while cumulative minutes were observed to be lower at follow-up, median scores remained static (T0: 2725 minutes, median 120; T1: 2055 minutes, median 120). In accordance with A-IPAQ instructions, if participants omitted either the time or days, then the case was removed from analysis [56].

There was a significant increase in mean (SD) total ESE-A scores from T0 597.5 (456.6) to T1 796.7 (564.3), z = -3.42, p < 0.001, indicating higher self-efficacy for exercise post-intervention compared to baseline, with a moderate to large effect size [r = 0.47 (95%CI 0.28 to 0.67)]. There was a significant decrease in BBAQ scores from T0 to T1 for one external barrier (social influence, p = 0.007) and two external barriers to exercise (fear of injury, p < 0.001; and lack of skill, p = 0.012), both with moderate to large effect sizes [r = 0.57 (95%CI 0.35 to 0.79) and r = 0.42 (95%CI 0.20 to 0.65), respectively]. Additionally, three environmental barriers were viewed as specific to the environmental context of Saudi Arabia and were included in this questionnaire. The statements are shown below, and the responses were scored similarly to the BBAQ. These were statistically significantly associated with a change in BBAQ scores from T0 to T1, each with a moderate effect size (Table 5).

The weather (extreme heat or cold) does not allow me to…

Physical activity in the summer is not suitable for me…

The surrounding environment does not allow me to engage in physical activity…

**Table 4. Pre-post comparisons for A-IPAQ and ESE-A.**

| A-IPAQ | Group | N | Sum of Ranks | Mean Rank | Z-value | Sig. | Effect size r (95%CI) | T0 Median METS (IQR) | T1 Median METS (IQR) |
|---|---|---|---|---|---|---|---|---|---|
| MET-Vigorous Physical Activity | Positive Ranks | 1 | 11 | 115 | -3.62 | <0.001 | 0.79 (0.54-1.04) | 0 (0-1920) | 2880 (960-5760) |
| | Negative Ranks | 19 | 219 | 115 | | | | | |
| | Ties | 1 | 1 | 1 | | | | | |
| MET-Moderate Physical Activity | Positive Ranks | 1 | 1.5 | 85.5 | -3.66 | <0.001 | 0.86 (0.60-1.12) | 80 (0-960) | 1440 (360-2400) |
| | Negative Ranks | 17 | 169.5 | 85.5 | | | | | |
| | Ties | 0 | 0 | 0 | | | | | |
| MET-Walking | Positive Ranks | 7 | 44.5 | 76.5 | -1.52 | 0.1297 | | 445.5 (0-2079) | 792 (264-1782) |
| | Negative Ranks | 10 | 108.5 | 76.5 | | | | | |
| | Ties | 0 | 0 | 0 | | | | | |
| The amount of weekday sitting (minutes per day) | Positive Ranks | 3 | 11 | 7.5 | 0.94 | 0.3452 | | 120 mins (60-360) | 120 mins (60-240) |
| | Negative Ranks | 2 | 4 | 7.5 | | | | | |
| | Ties | 0 | 0 | 0 | | | | | |
| **ESE-A** | **Group** | **N** | **Sum of Ranks** | **Mean Rank** | **Z-value** | **Sig.** | | | |
| Total | Positive Ranks | 10 | 293 | 667 | -3.42 | <0.001 | 0.47 (0.28-0.67) | | |
| | Negative Ranks | 33 | 1041 | 667 | | | | | |
| | Ties | 9 | 45 | 45 | | | | | |

A-IPAQ: Arabic International Physical Activity Questionnaire. ESE-A: Arabic version of Exercise Self-Efficacy scale MET: Metabolic Equivalent Task. Data were from 39 participants. Effect size (r) was calculated as rank-biserial correlation. T0-baseline; T1- follow up.

## Nested qualitative study

Qualitative data were collected by the study researcher from a convenience sample of patients and nurses involved in their care. This was part of intervention fidelity assessment, and the purpose/ anticipated response is outlined in Table 2. Reporting is guided by the Consolidated Criteria for Reporting Qualitative Research (COREQ-32) [62] (S4 Table). The study team developed semi-structured interview guides (one for patients and one for nurses) (S5 Table). Nineteen individuals took part in the qualitative study. Interview participant characteristics are shown in Table 6. Eleven participants were patients with T2DM (7 male, 4 female), and eight healthcare participants who were nurses involved in their clinical care (all female). The mean age of patients was 54.54 years (SD = 12.5 range 36–69 years). The mean age of the nurses was 31.75 years (SD = 3.0 range 27–36 years).

Interviews took place during the week following the end of the intervention (week 7) and were conducted by the study researcher (MJA) over the telephone. Questioning explored participants' views towards the ActiveText@T2D text messaging intervention, as well as patients' level of engagement with message content and behavioural recommendations and any barriers or enablers of engagement with the intervention or trial processes. Several strategies were adopted to enhance rigour, including building rapport with participants in the clinic setting, active use of researcher reflexivity, rigorous peer review by a wider study team, and triangulation of qualitative findings with the quantitative data and peer reflections. These were brief interviews lasting on average 6.5 minutes. Interviews were audio-recorded with consent, transcribed verbatim in the Arabic language, and then translated into English by the researcher. Back translation was undertaken by a professional translator (fluent in English and Arabic) and compared to the original transcripts for inconsistencies. Data were analysed using inductive thematic analysis [63] supported by the NVivo software system (version 12). This comprised six phases: familiarisation with the data; generation of initial codes; searching for themes; reviewing themes; defining and naming themes in a code book; and producing the report. Three themes and 13 sub-themes were identified (Fig 2). Illustrative quotations are provided in Table 7.

**Table 5. Pre- post comparison of BBAQ scores.**

| Barriers to Physical Activity | Group | N | Sum of Ranks | Mean Rank | Z-value | Sig. | Effect size r (95%CI) |
|---|---|---|---|---|---|---|---|
| Lack of Time | Positive Ranks | 13 | 326.5 | 362.5 | -0.51 | 0.611 | |
| | Negative Ranks | 16 | 398.5 | 362.5 | | | |
| | Ties | 10 | 55 | 55 | | | |
| Social Influence | Positive Ranks | 7 | 167.5 | 352.5 | -2.71 | 0.007 | 0.44 (0.22-0.66) |
| | Negative Ranks | 23 | 537.5 | 352.5 | | | |
| | Ties | 8 | 36 | 36 | | | |
| Lack of Energy | Positive Ranks | 13 | 290 | 328 | -0.60 | 0.548 | |
| | Negative Ranks | 19 | 366 | 328 | | | |
| | Ties | 4 | 10 | 10 | | | |
| Lack of Willpower | Positive Ranks | 19 | 430 | 346.5 | 1.27 | 0.206 | |
| | Negative Ranks | 14 | 263 | 346.5 | | | |
| | Ties | 4 | 10 | 10 | | | |
| Fear of Injury | Positive Ranks | 7 | 116 | 344 | -3.46 | 0.001 | 0.57 (0.35-0.79) |
| | Negative Ranks | 25 | 572 | 344 | | | |
| | Ties | 5 | 15 | 15 | | | |
| Lack of Skill | Positive Ranks | 7 | 149.5 | 301 | -2.50 | 0.012 | 0.42 (0.20-0.65) |
| | Negative Ranks | 21 | 452.5 | 301 | | | |
| | Ties | 7 | 28 | 28 | | | |
| Lack of Resources | Positive Ranks | 21 | 422 | 348.5 | 1.12 | 0.265 | |
| | Negative Ranks | 13 | 275 | 348.5 | | | |
| | Ties | 3 | 6 | 6 | | | |
| Temperature extremes do not allow me to exercise* | Positive Ranks | 24 | 594 | 337.5 | 3.80 | <0.001 | 0.62 (0.40-0.83) |
| | Negative Ranks | 3 | 81 | 337.5 | | | |
| | Ties | 11 | 66 | 66 | | | |
| Physical activity in the summer is not suitable for me * | Positive Ranks | 8 | 195 | 331.5 | -2.03 | 0.042 | 0.33 (0.11-0.55) |
| | Negative Ranks | 18 | 468 | 331.5 | | | |
| | Ties | 12 | 78 | 78 | | | |
| The surrounding environment does not allow me to engage in physical activity* | Positive Ranks | 8 | 194 | 344.5 | -2.15 | 0.031 | 0.34 (0.13-0.56) |
| | Negative Ranks | 18 | 495 | 344.5 | | | |
| | Ties | 13 | 91 | 91 | | | |

*These three environmental factors are not part of the original seven BBAQ domains. Data were from 39 participants. Effect size (r) was calculated as rank-biserial correlation.

**Theme 1: Use of text messaging as a health intervention.** Participants with T2DM spoke of the importance of trust in providers of health messages. This text messaging intervention provided a trusted source of messaging, since participant recruitment had taken place in the hospital setting, and therefore individuals were aware that messages were associated with a healthcare provider. Participants perceived the messages to be easy to access, appropriate for the Saudi population, and flexible in that they were able to open and read them at a convenient time. The timing of delivery was seen to be acceptable and technical challenges were minimal, although one participant experienced difficulties with access due to weak network signals and another ignored some messages due to a change in service provider. For those who lived outside of the capital city, particularly in rural areas, the messaging was viewed to be a valuable approach to sustaining a connection between the healthcare recipient and the healthcare provider. However, there were some

**Table 6. Interview participant characteristics.**

**Patients' characteristics**

| Participant ID | Gender | Age | Occupation |
|---|---|---|---|
| Patient (07) | Male | 38 | Salesman |
| Patient (13) | Male | 58 | Retired (professional) |
| Patient (22) | Male | 36 | Engineer |
| Patient (23) | Female | 69 | Social media influencer |
| Patient (28) | Female | 66 | Housewife |
| Patient (29) | Male | 64 | Researcher |
| Patient (31) | Male | 65 | Trader |
| Patient (40) | Female | 39 | Housewife |
| Patient (42) | Male | 59 | Retired (manual) |
| Patient (55) | Female | 61 | Housewife |
| Patient (60) | Male | 45 | Retired (manual) |

**Nurses' characteristics**

| Participant ID | Gender | Age | Occupation |
|---|---|---|---|
| Nurse (1) | Female | 34 | Ward nurse |
| Nurse (2) | Female | 29 | Nurse educator |
| Nurse (3) | Female | 30 | Charge nurse |
| Nurse (4) | Female | 36 | Head nurse |
| Nurse (5) | Female | 33 | Ward nurse |
| Nurse (6) | Female | 27 | Ward nurse |
| Nurse (7) | Female | 34 | Nurse administrator |
| Nurse (8) | Female | 31 | Ward nurse |

concerns that messaging may not suit all patients with T2DM, for example, some older adults, or those with low health literacy.

**Theme 2: Engagement with physical activity.** For participants with T2DM, the text messages served as useful reminders of the importance of physical activity for diabetes control, and the motivation to adhere to physical activity advice. For most, the messaging led to actions of behavioural intention (e.g., enrolling at a gym) and/or behavioural change (i.e., increased physical activity). The mechanisms of action primarily included (1) building participants' confidence for physical activity (e.g., for certain sub-groups such as older adults, in situations where it was more challenging, or alongside other aspects of T2DM management), and (2) increasing participants' perception of social support for lifestyle changes (e.g., from the research or clinical team, or from friends and family). For individuals who had not observed changes in their diabetes resulting from physical activity, there was an increased awareness that physical activity engagement would need to be sustained over time to reap benefits for their health. Some participants with T2DM experienced barriers to physical activity engagement. These included individual factors (i.e., lack of motivation), competing demands on their time, caregiving responsibilities, gender issues (i.e., social and cultural expectations for women), environmental conditions (i.e., hot weather) and a lack of social support. For nurses, the intervention was seen to be a particularly valuable adjunct to clinical care in a developing country where the structure of T2DM services can be sub-optimal, with limited or no support available for patients to make lifestyle changes.

**Theme 3: Instilling knowledge about physical activity and diabetes control.** There was a general lack of awareness among participants of the importance of lifestyle behaviours in the management of T2DM. Therefore, the messages provided new knowledge for most, not only for patients, but also the nurses involved in their clinical care. It seemed that physical activity was not generally included in discussions between clinical staff and their patients, and

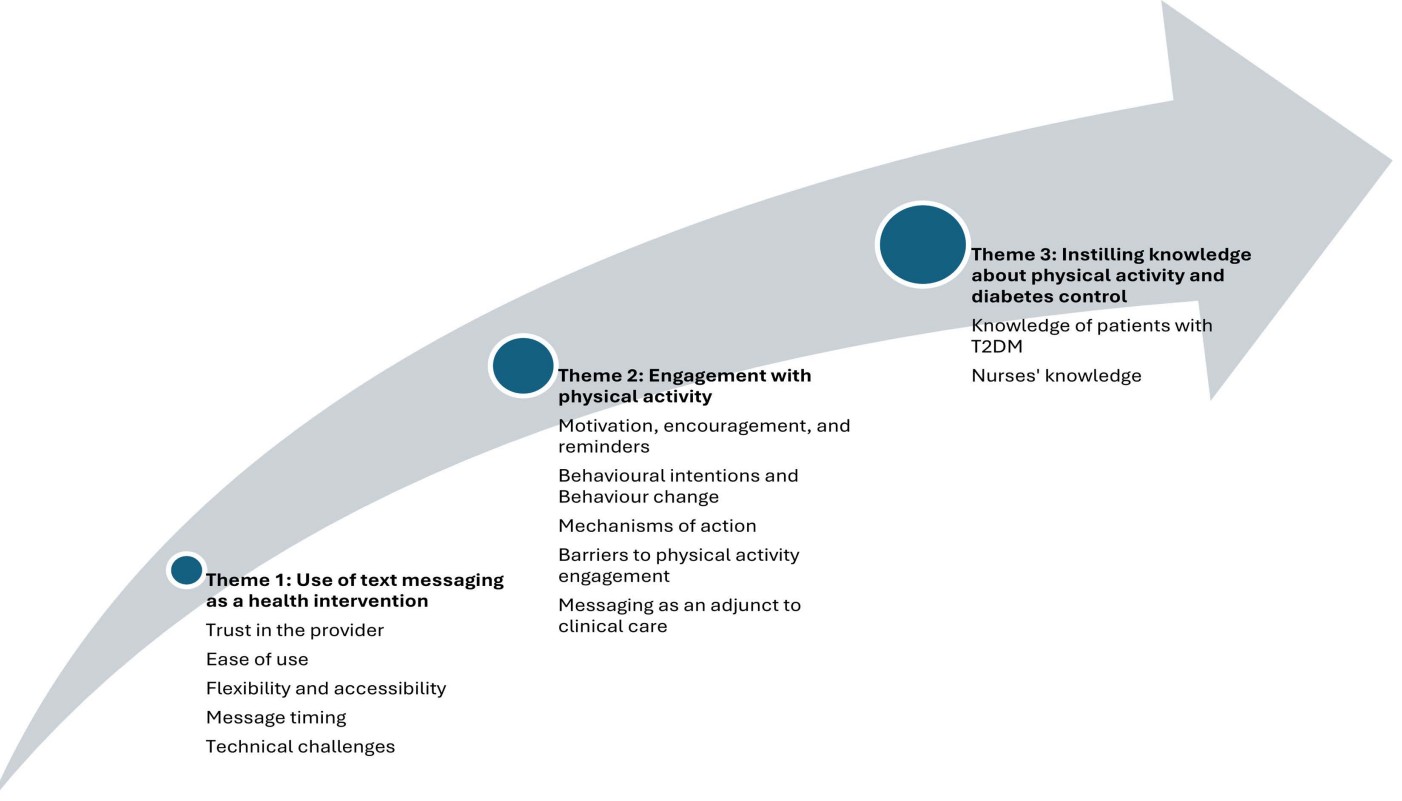

**Fig 2. Key themes and sub-themes.**

this new knowledge resulted in nurses intending to change their health advice to patients to include a greater focus on physical activity. Importantly, some nurses reported that they would be making changes to their own physical activity levels due to the intervention (including one nurse who was pre-diabetic). It was suggested that health education about the role of lifestyle intervention in T2DM was needed for Saudi healthcare professionals, as well as their patients.

## Discussion

To our knowledge, this was the first study to develop and test a culturally tailored text messaging behavioural intervention aimed at increasing physical activity in Saudi adults with T2DM. The ActiveText@T2D intervention was found to be feasible and acceptable to patients with T2DM and healthcare professionals involved in their care. Initial interest in the intervention was high, with most eligible participants consenting to take part in the study. Our recruitment rate (84% of those eligible) was significantly higher than that reported in other studies involving text messaging interventions for diabetes management (11–47%: [64–67]). Study retention was high; baseline measures were completed by 85%, with 75% retention to follow-up, a higher retention rate than reported in other text messaging intervention studies [68,69]. The level of burden for baseline and follow-up surveys was deemed to be acceptable to participants given the high retention to follow-up, missing data were minimal and there were no reported adverse events.

Although comparison of pre- and post- data is not the focus of a feasibility study, scores on our outcome measures from T0 to T1 indicated that the selected measures were sensitive to change following the ActiveText@T2D intervention, and therefore suitable for use in a future randomised trial. Our study suggests that the ActiveText@T2D intervention may increase physical activity, specifically, MET Moderate Activity and MET Vigorous Activity. While there were no statistically

PLOS Digital Health

**Table 7.  Themes and illustrative quotations.**

| Theme | Illustrative quotation |
|---|---|
| Theme 1: Use of texts as a health intervention | |
| Trust in the provider | "I worry a lot about getting the correct information and education about my diabetes. …I trusted your messages because you are working here". (Patient 60)<br>"They are useful if the sender's name is clear, or I have them in my phone contacts". (Patient 42) |
| Ease of use | "They are easy and fast". (Patient 31)<br>"…they seemed easy to understand". (Nurse 8)<br>"I think it is suitable for Saudis, as people here are familiar with SMS messages, they like them as we used them for appointments reminders, and we saw it is effective and easy to open and they told us it is good". (Nurse 1) |
| Flexibility and accessibility | "…particularly for those living in far cities not close to big cities like Riyadh". (Patient 29)<br>"…this is a very important social service". (Patient 23)<br>"SMS messages are sometimes better than normal conversations with patients. They are clearer and easier…" "SMS is a good method to be used for both contacting or being contacted". (Nurse 4)<br>"It depends on their age, some people don't know how to read, or they prefer face-to-face meetings". (Patient 22)<br>"Yes, for sure using a mobile is important and it is a big part of everybody's lives, but maybe some people cannot read or use the phone". (Patient 60)<br>"Probably, you need to tackle more communication options with the patients instead of only the text messaging tool". (Nurse 8) |
| Message timing | "I received them between the Maghrib and Isha prayer times while I was taking my routine coffee". (Patient 40)<br>"As long as you have patients' numbers, it is always easy to stay in touch with them and they can read your messages anytime". (Nurse 4) |
| Technical challenges | "Not sure how many messages I received, I live outside Riyadh, and sometimes the network is weak, so not sure". (Patient 55)<br>"In recent weeks, the title of messages changed, I mean the service provider, which made me ignore some of them". (Patient 29) |
| Theme 2: Engagement with physical activity | |
| Motivation and encouragement to engage | "Thank you for the motivational messages, I received them, and they encouraged me a lot to walk". (Patient 29) |
| Behavioural intentions (texts serving as reminders of the importance of PA) | "I enrolled at the gym and hope to continue with that". (Patient 42)<br>"The messages were helpful, they changed my thinking" (Patient 22). |
| Behaviour change (texts prompting action) | "I started using stairs more than before, I also started walking, one time I walked more than five thousand steps". (Patient 22)<br>"I like walking, it makes me feel good and healthy". (Patient 07)<br>"Not a remarkable change, but inshallah this may be seen in the long run". (Patient 29) |
| Mechanisms of action (texts improving confidence, and encouraging social support/ message sharing) | "It was very motivating and reminded me when I felt lazy, I felt more courage to do any kind of exercise even if it was not easy". (Patient 31)<br>"The messages helped me especially when I was busy or not paying attention, I feel you challenged me". (Patient 28)<br>"…these messages made me feel as if there was someone supporting and encouraging me" (Patient 13).<br>"For me, any word of support is like medicine …when you sent me those messages, it means you care about me". (Patient 23)<br>"I shared them with my older sister so she could help herself to walk more". (Patient 13) |
| Barriers to physical activity engagement | "I love exercise, but my social life is a big barrier all the time. I take care of my kids and husband, and even his family; they all take a lot of my time". (Patient 55)<br>"It is not easy when you live with people that do not support you". (Patient 40)<br>"…our weather is so hot and miserable". (Patient 13)<br>"I love exercise, but I always forget and feel lazy". (Patient 31). |
| Messaging as an adjunct to clinical care | "Patients can easily forget about their treatment management plan and SMS reminders can fill that gap". (Nurse 3)<br>"…here most of patients do not want to take any encouragement or initiatives to modify their lifestyles, so your study is a good initiative…They need outside encouragement like this…we need some initiatives to give some instructions to patients, instead of only taking information or vital signs" (Nurse 1) |

*(Continued)*

**Table 7.** (Continued)

| Theme | Illustrative quotation |
|---|---|
| Theme 3: Texts instilling knowledge about physical activity and diabetes control | |
| Patients' acquisition of new knowledge | "I learned about the importance of exercise on my diabetes control, inshallah, I will be active every day". (Patient 31) |
| Nurses' acquisition of new knowledge | "I was thinking only medication can improve the patient's diabetic condition. But I know now physical activity is also important. Patients must do some exercise to improve their blood sugar". (Nurse 2)<br>"I have gained knowledge which makes me want to apply it to myself first, because I am overweight and have prediabetes" (Nurse 6)<br>"I have learnt we should add physical activity to our routine treatment not only medication or vital signs. Our doctors do not tell us about these things, I do not remember if we told patients about physical activity". (Nurse 5)<br>"I think we need to learn as well; it is important for all of us". (Nurse 3) |

significant differences in weekday sitting time and MET Walking, given the exploratory nature of analysis in a feasibility study, we advocate that these variables are further explored in a large-scale definitive trial.

We also found that the intervention could reduce some barriers to exercise in Saudi patients with T2DM. Self-efficacy, a determinant of physical activity behaviour change, increased post-intervention, which has been found in text messaging interventions with T2DM patients in other settings (e.g., Iran: [70]). The significant changes in outcomes observed in this study demonstrates that the measures used are sensitive to change and could be used in a definitive trial. Since this is a feasibility study, all statistical analyses are exploratory and therefore statistically significant results should not be over-interpreted due to limited power.

The ActiveText@T2D intervention had high fidelity. Most participants received messages as planned, with only a small number of messages undelivered due to unavoidable technical issues (i.e., mobile phone signal). The messaging schedule (timing, frequency) was found to be acceptable to most participants. Our nested qualitative study provides insights into perceptions towards, and acceptability of, the intervention. Primarily, the intervention was positively perceived, by patients and healthcare professionals, as a valuable adjunct to clinical care. Particularly valued aspects of the intervention were its ease of use, flexibility, and accessibility via mobile phone.

Trust in the provider was an important enabler of study recruitment and intervention engagement; this aligns with prior reports that establishing trust in the research team is a key recruitment challenge for intervention studies [71]. Patients referred to feeling cared for and supported due to the intervention, which aligns with findings from other studies focused on diabetes care in which text messaging has functioned as 'psychosocial support' [72]. Similarly, our qualitative data suggest that the ActiveText@T2D intervention has the potential for broader community benefit since the messages encouraged discussion about physical activity between patients and their friends and/or family. In some instances, this led to family members (without T2DM) becoming more active.

Patients with T2DM and healthcare professionals reported the acquisition of new knowledge following the intervention. While discussion about physical activity occurred as a direct result of the ActiveText@T2D intervention, our qualitative findings highlighted that physical activity is rarely discussed in clinical consultations between patients with T2DM and their healthcare providers. This intervention was perceived to provide new knowledge for its recipients, as well as the motivation and confidence to make, and sustain, behavioural changes, capitalise on enablers of physical activity and address any key barriers to active lifestyles. While this supports the delivery of the ActiveText@T2D intervention alongside standard clinical care, the reported absence of any discussion related to health behaviour modification in patients' clinical consultations has broader implications for T2DM management. A key component of which is education, self-management, and lifestyle modification, including engagement in physical activity [73]. It is notable that nurses described a focus on clinical indicators of diabetes control (e.g., HbA1c/ vital signs) and medication, but an absence of knowledge (or discussion)

relating to health behaviours, such as physical activity. This highlights an urgent need to incorporate education relating to T2DM and health behaviours into continuing professional development training for healthcare professionals in Saudi Arabia. Similarly, a review of content related to T2DM management within curricula for trainee healthcare professionals could be warranted.

Survey research conducted in the Middle East has suggested that patients with T2DM are willing to use mobile phones to support their diabetes management (Iran: [74]), which supports our overall rationale. In developing countries, the penetration rate of smartphones is lower than text message usage, which would restrict the impact of more advanced mHealth approaches - basic phone usage and text messaging remain important communication methods. Therefore, text messaging remains a novel and valuable approach to diabetes in this region, given its potential for high reach and scalability. Our text messages were theory-based, using the COM-B, which covers a broader range of influences on behaviour than many older theoretical models. The COM-B is commonly used in more recent research to guide the design of text messaging interventions in a range of clinical populations and settings (e.g., [75–77]).

There are few studies using text messaging in adult diabetes care in the Middle East region, and to our knowledge, there are no other studies that have adequately described the intervention (development and content) and focused specifically on the reporting of feasibility and acceptability outcomes. Systematic reviews of text messaging interventions in diabetes care include studies focused on the 'effectiveness' of interventions (e.g., in improving glycaemic control) rather than 'feasibility and acceptability' [37,78] and highlight the paucity of text messaging studies conducted in the Middle East. A small number of diabetes text messaging interventions have been conducted in the Middle East region, including Iran, Iraq, Jordan, Türkiye, Egypt and Saudi Arabia. Our study focuses solely on physical activity in the T2DM population in Saudi Arabia. It differs to previously published research for several methodological reasons. A feasibility trial as defined by the MRC framework for evaluating complex interventions [79] uses pre-determined criteria for "reducing uncertainty around recruitment, data collection, retention, outcomes, and analysis) or the intervention itself (e.g., around optimal content and delivery, acceptability, adherence…)". However, prior studies (i) use text messaging as an adjunct to another intervention type (e.g., text reminders following group education in Türkiye [80] and Saudi Arabia [81]), (ii) focus on specific groups only (e.g., pre-diabetic pregnant women in Iran [82]) or specific diabetes-related complications or treatment (e.g., diabetic foot care in Iran [83] and Jordan [84], or insulin injection technique in Türkiye [85]), (iii) did not promote physical activity (Iraq:[86]; Jordan [84]) or measure physical activity-related outcomes (Egypt [87]), (iv) focused on effectiveness not feasibility outcomes (Iran [70,83,88,89] and Saudi Arabia [90]), (v) and/or the description of the SMS intervention is either missing (Iran [70]) or very limited (Iran [89,91], Jordan [84], Saudi Arabia [81,90] and Türkiye [80]). One early paper with a diabetes text messaging intervention was retracted (Iran: [92]). Further, the geographical regions in which prior research was conducted - Iran, Iraq, Jordan, Türkiye, Egypt, Saudi Arabia - while sharing commonalities (e.g., region, religion) have distinct cultural differences which are likely to impact on physical activity engagement and therefore messaging content. These differences are influenced by various factors, including climate, lifestyle, infrastructure, gender norms, and government initiatives.

## Study limitations

Most of the eligible participants present in the clinic during the recruitment period were recruited into the study. Since the recruited sample is predominantly moderate-to-high income, well educated, and from an urban setting, this limits generalisability to more rural, lower-literacy, or socioeconomically disadvantaged Saudi populations. The number and characteristics of those who failed to attend their scheduled clinic appointment or did not have a scheduled appointment during the recruitment period, are not known. Therefore, we are unable to compare characteristics of our recruited sample with the wider pool of potentially eligible participants with T2DM. It is possible that those who do not attend may have specific characteristics, such as better diabetes control, lower educational level, or poor employment status (i.e., lower income), all of which have been associated with non-attendance at T2DM outpatient appointments [93].

The intervention delivery and data collection occurred in 2020. Although this was several years ago, the findings are still highly relevant. This is because the prevalence of T2DM remains high and is increasing in Saudi Arabia [9–11,13], meaning that efforts to engage the population in physical activity remain critical. There are currently no comparable theory-based interventions that use text messaging to promote physical activity in people with T2DM in Saudi Arabia. There are no other well-designed feasibility studies reporting on the development, feasibility and acceptability of similar diabetes text messaging interventions, and therefore our intervention and findings remain novel.

The intervention was based solely on SMS text messaging; although this met the needs of our participants in 2020, digital health technologies have evolved in the interim and it is possible this could impact current feasibility or generalisability. Nonetheless, the paucity of studies reporting on text messaging interventions in developing countries, and the continued prevalence of text messaging as a form of communication, means that, as stated earlier and confirmed through our PPIE work, text messaging-based interventions remain an appropriate and novel health promotion approach in the context of Saudi Arabia.

SMS text messaging may be viewed as limited in other countries and contexts in a time of rapidly advancing communication technologies. However, text messaging interventions have wide reach and have shown to engage and benefit patients with T2D, regardless of health literacy status [94]. Further, prior research shows that adding components to text-only interventions does not improve their efficacy [95]. The topic area is limited to physical activity, and future studies may wish to consider adding other important aspects of diabetes self-management to the messaging (e.g., diet, medicines adherence).

Due to limitations of time and resources, our nested qualitative study is limited to a small sub-sample and the interviews were brief, although the qualitative work was sufficient to meet our feasibility study objectives and identified important issues. Our strategies to enhance recruitment have been used previously [96–98]. A few participants reported that the questionnaires were too long and time-consuming. This may reflect burden of data collection measures for a small number of participants, and burden is known to affect response rates and data quality [99]. However, most participants did not experience burden. Indeed, participants reported that they valued using their clinic waiting time productively, which is consistent with previous studies [100]. It may be that some participants had insufficient time to read the participant information and complete baseline measures in the clinic setting. This should be considered in the design of a future trial. Although we endeavoured to use short versions of measures, since short versions can lead to higher responses [80], respondents might require a longer time to read the study information and complete questionnaires [101]. Some patients declined due to concerns about confidentiality relating to sharing their health status in a public environment (i.e., the clinic waiting room) or in the presence of nurses or family members, which has been found in previous studies [102]. Future studies might consider the venue for data collection and ideally provide a more private area to address any confidentiality concerns within the recruitment process [102,103]. We did not use objective measures of physical activity, so findings are limited to self-reports, in which physical activity levels can be over-estimated [104,105]. The IPAQ should be interpreted with caution in those with lower literacy [52]. Finally, while this study achieved its aims and objectives with relation to ascertaining feasibility and acceptability, the comparison of outcome measures pre and post intervention (albeit not the primary focus of a feasibility study) is based on a small sample, and pre-post comparisons of measures are therefore exploratory with limited generalisability. Nonetheless, our sample size was appropriate for the study design since a review of sample sizes for feasibility studies [106] shows that, although there is variability, the average number of participants is 36 patients per study arm (IQR 25–50). This is a single arm feasibility trial with 52 participants and therefore our sample is relatively large compared to the average for this type of study.

## Future directions

Additional insights could be gathered by conducting in-depth qualitative interviews or focus group discussions. Ideally of a longer duration and a larger sample of diabetes patients who have used ActiveText@T2D, and healthcare professionals

involved in their care. The views of other healthcare professionals involved in diabetes care would be valuable (e.g., physicians). Further research could explore which of the messages were more impactful, why, and to whom. Following this feasibility study, a future definitive randomised controlled trial would allow for the assessment of outcome 'effectiveness' and could incorporate objective measures of physical activity, assessment of intervention cost-effectiveness, and a process evaluation to explore intervention implementation across a range of contexts and settings. This study demonstrates the acceptability of text messaging interventions in this context which could be developed to address other health behaviours and areas of self-management or applied to other health conditions.

## Conclusion

To our best knowledge this is the first study to explore the experience and attitude of participants (patients with T2DM) and healthcare professionals (nurses) towards the use of text messages in the context of Saudi Arabia. Our findings confirm the feasibility and acceptability of the ActiveText@T2D intervention which could be tested in a definitive trial to determine the effectiveness and cost-effectiveness of this intervention. This research builds on prior studies conducted in other cultural contexts by continuing to explore the use of text messages to promote healthier lifestyle intervention programmes in adult diabetes populations [78,107,108], and confirms the value of using a unique, culturally tailored intervention. Should the ActiveText@T2D intervention show to be effective and cost effective, it could be included in national diabetes programs to support self-management of T2DM, integrated with existing healthcare workflows in secondary and primary care, and aligned with broader digital health strategies in Saudi Arabia and other Gulf countries, such as Vision 2030 [109].

## Supporting information

**S1 Table. Text messages content mapped to BCW and COM-B.**
(DOCX)

**S2 Table. CONSORT 2010 checklist.**
(DOC)

**S3 Table. The TIDieR checklist.**
(DOCX)

**S4 Table. COREQ-32 checklist.**
(DOCX)

**S5 Table.** Interview topic guides.
(DOCX)

**S1 Checklist. Inclusivity in global research questionnaire.**
(DOCX)

## Acknowledgments

The authors thank all those who supported approvals processes, site access, administration, informed consent and enrolment of patients to the study. In particular: Dr. Abdul-Aziz Alhumaidy, Dr. Mohammed Al-Mohaithef, Dr. Basma Bouqs, Dr. Ibrahim Al Qassimi and Mr. Muath. Dr Andrea Venn is thanked for providing statistical consultation.

## Author contributions

**Conceptualization:** Holly Blake, Mohammed Jamaan Alsahli.

**Data curation:** Mohammed Jamaan Alsahli.

**Formal analysis:** Holly Blake, Mohammed Jamaan Alsahli, Wendy J. Chaplin.

**Funding acquisition:** Mohammed Jamaan Alsahli.

**Investigation:** Mohammed Jamaan Alsahli.

**Methodology:** Holly Blake, Stathis Th. Konstantinidis.

**Project administration:** Mohammed Jamaan Alsahli, Wendy J. Chaplin.

**Resources:** Holly Blake.

**Supervision:** Holly Blake, Stathis Th. Konstantinidis.

**Writing – original draft:** Holly Blake, Mohammed Jamaan Alsahli.

**Writing – review & editing:** Wendy J. Chaplin, Stathis Th. Konstantinidis.

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
