## [Decision Letter · Decision Letter 0]

Response to Reviewers Revised Manuscript with Track Changes Manuscript
**Journal Requirements:**

**Additional Editor Comments (if provided):**
**Reviewers' Comments:**

**Comments to the Author**

1. Does this manuscript meet PLOS Digital Health’s publication criteria?

Reviewer #1: Yes

Reviewer #2: Partly

Reviewer #3: Partly

2. Has the statistical analysis been performed appropriately and rigorously?

Reviewer #1: Yes

Reviewer #2: No

Reviewer #3: Yes

3. Have the authors made all data underlying the findings in their manuscript fully available (please refer to the Data Availability Statement at the start of the manuscript PDF file)?

Reviewer #1: Yes

Reviewer #2: Yes

Reviewer #3: Yes

4. Is the manuscript presented in an intelligible fashion and written in standard English?

Reviewer #1: Yes

Reviewer #2: Yes

Reviewer #3: Yes

Reviewer #1: This is a manuscript on an important topical issue (T2DM) carried out among a population living in a challenging environment that does not favor favour outdoor physical activity; and certainly for women, cultural barriers are present. SMS text messaging is a tool that has the potential to reach many at their convenience.

The paper is well presented and meticulously structured and supported by current literature that is well cited and referenced.

I have highlighted few areas - might be typos - for the authors to revisit:

My copy has line numbers missing from page 14 - so I will use page numbers:

1- Page 20, is the second "for" before "Total MET" necessary? -For A-IPAQ, for total MET (Metabolic Equivalent Task: the ratio of the rate of energy expended during an activity to the rate of energy expended at rest), MET Walking, MET Moderate Activity and MET Vigorous Activity all increased from T0 to T1

2. Page 26 - Figure 4, theme 2 - pleae confirm that abbreviation "PA" has been defined appropriately

3. Page 29 - Category - Nurses acquisition of new knowledge - I was thinking only medication can improve the patient diabetic condition. Would you want to use "patient's'? or as reported (sic)?

4. Finally, page 68 - Confirm PLOS DH guidelines on whether "Conclusion" is more appropriate than "Summary".

Otherwise a clean manuscript!

Reviewer #2: This article presents a single-arm feasibility trial of a culturally tailored text messaging (SMS) intervention—ActiveText@T2D—designed to increase physical activity in Saudi adults with type 2 diabetes (T2DM). Over a 6-week intervention, participants received two text messages per week, with the content informed by the Behaviour Change Wheel (BCW) and COM-B model. Feasibility endpoints included recruitment, retention, fidelity, and acceptability. Quantitative measures (IPAQ, ESE-A, BBAQ, and clinical data) and nested qualitative interviews explored perceived benefits and barriers. The authors emphasize the cultural adaptation of messages, the inclusion of expert panel review, and a patient public involvement component. They conclude that ActiveText@T2D is acceptable and feasible in this population, meriting further investigation in a larger, definitive trial.

--- Pros

- The introduction provides enough epidemiological context on the burden of T2DM globally and in Saudi Arabia, highlighting the importance of physical activity in diabetes management.

- It clearly identifies a research gap: very few text-based interventions specifically target physical activity in Saudi T2DM populations.

- An expert panel review and PPIE input were used to validate message content and ensure cultural appropriateness.

- A nested qualitative study adds depth to the results by exploring participant and healthcare professional perspectives on the intervention.

- The study addresses a regional gap—there are few culturally tailored, SMS-focused interventions targeting physical activity in Saudi adults with T2DM. The high prevalence of T2DM in Saudi Arabia adds practical significance to these findings.

--- Cons

- The defined objectives are not significant enough.

- The sample size is too small, 52 patients (not good enough).

- The experiment ran 5 years ago (not very recent).

- Although 6 weeks is presumably based on habit-formation timelines, the rationale remains vague. Explaining precisely why six weeks, and why specifically two messages per week, would strengthen the methodological justification.

- Methods for dealing with incomplete questionnaires are not fully detailed: how many incomplete records might bias results.

- Clarification needed regarding sample size decisions, missing data approaches, and strategies for dealing with the bias of a predominantly well-educated, higher-income cohort.

- The recruited sample is predominantly moderate-to-high income, well educated, and from an urban setting. This limits generalizability to more rural, lower-literacy, or socioeconomically disadvantaged Saudi populations.

- The discussion acknowledges generalizability issues but does not elaborate enough on how different demographics might have distinct feasibility barriers.

- The results are not well-articulated.

- Deepen the discussion of how future trials would incorporate objective measures, randomized design, and cost-effectiveness or scale-up analysis.

- Strengthen the qualitative component or justify its brevity, considering whether more in-depth interviews or focus groups could yield richer insights.

--- Decision

I believe this manuscript does not meet the high standards required for publication in this journal.

Reviewer #3: Dear Authors,

Thank you for the opportunity to review your manuscript. Overall, this study is well-conceived, well-written, and presents an important and relevant contribution to the field of mHealth and type 2 diabetes management. The use of a culturally tailored SMS intervention for increasing physical activity in Saudi adults with T2DM is novel, and your work is comprehensive in its approach.

Below are a few comments that may help improve the clarity, rigor, and adherence to PLOS guidelines.

Abstract:

I believe the abstract could include more about the numeric outcomes, despite being a feasibility study. For example, adding more information about the participants as well as the self-reported outcomes. Additionally, the discussion part of the abstract should be expanded with more about the next steps.

Introduction:

The introduction provides a solid background and literature review. However, in the first paragraph, the discussion of type 2 diabetes globally is followed by references that primarily focus on Saudi Arabia. Consider adjusting to ensure coherence.

Minor editing issues are present in lines 92 and 102 that need attention.

Methods:

The manuscript should provide a more detailed description of the statistical methods used, including the specific significance threshold (e.g., p < 0.05). This transparency is essential for reproducibility and aligns with PLOS's guidelines.

Line 214 (American Diabetes Association) requires a citation.

Results:

•The manuscript mentions changes in MET walking and sitting time, but statistical significance is inconsistent across text and tables. For example:

oThe text suggests significance for MET walking, but the data indicate otherwise.

oThe same discrepancy is present for sitting time.

•The reporting of outcomes should include effect sizes or percentage changes to provide better context. For example, when stating “an increase in MET scores,” it would be useful to specify by how much.

•The qualitative claims regarding text messages serving as reminders (e.g., "the text messages served as useful reminders of the importance of physical activity for diabetes control") should be backed by more direct evidence from the interviews.

•Figure 3 effectively conveys the necessary information, while Figure 2 appears redundant. Consider removing Figure 2.

•The discussion in Page 27 (“The mechanisms of action…”) seems to repeat parts of the methods section and could be reduced.

•Further clarity on the nurse interviews is needed—why were they conducted, and what were the expected insights?

Discussion:

•Overall, I believe that the discussion would benefit from being expanded to integrate more findings from the results. Currently, it lacks attention to some of the results (like the self-reported outcomes) as well as more focus on the fact that this is a feasibility study (with its limitations).

•Also, a more in-depth discussion on the nutritional aspects of T2D would be beneficial, as currently it only mentions physical activity and medications as interventions for T2D.

•Ensure that the discussion maintains the feasibility focus, particularly by emphasizing that statistically significant results should not be overinterpreted due to limited power.

•Given that the study was conducted in 2020 and is being published in 2025, consider adding a note in the discussion about how digital health technologies have evolved in the interim and whether this could impact feasibility or generalizability.

•Please also expand the limitations section to include a discussion on generalizability and possible future directions.

General Comments:

The manuscript lacks line numbering from page 14 onward. Please ensure that line numbers are included on all pages.

Overall, the manuscript is well-structured and contributes meaningfully to the field of digital health and diabetes management. I hope these comments are useful in strengthening your manuscript.

Best regards,

**Do you want your identity to be public for this peer review?** For information about this choice, including consent withdrawal, please see our Privacy Policy

Reviewer #1: **Yes: ** Eric Mugambi Nturibi

Reviewer #2: No

Reviewer #3: **Yes: ** Shlomo Yeshurun

**Figure resubmission:****Reproducibility:** To enhance the reproducibility of your results, we recommend that authors of applicable studies deposit laboratory protocols in protocols.io, where a protocol can be assigned its own identifier (DOI) such that it can be cited independently in the future. Additionally, PLOS ONE offers an option to publish peer-reviewed clinical study protocols. Read more information on sharing protocols at https://plos.org/protocols?utm_medium=editorial-email&utm_source=authorletters&utm_campaign=protocols

---

## [Decision Letter · Decision Letter 1]

Response to Reviewers Revised Manuscript with Track Changes Manuscript
**Journal Requirements:**
**Additional Editor Comments (if provided):**
**Reviewers' Comments:**

**Comments to the Author**

Reviewer #2: (No Response)

Reviewer #3: (No Response)

Reviewer #4: All comments have been addressed

Reviewer #5: (No Response)

publication criteria?

Reviewer #2: No

Reviewer #3: (No Response)

Reviewer #4: Yes

Reviewer #5: Yes

3. Has the statistical analysis been performed appropriately and rigorously?

Reviewer #2: Yes

Reviewer #3: (No Response)

Reviewer #4: Yes

Reviewer #5: Yes

4. Have the authors made all data underlying the findings in their manuscript fully available (please refer to the Data Availability Statement at the start of the manuscript PDF file)?

Reviewer #2: Yes

Reviewer #3: (No Response)

Reviewer #4: Yes

Reviewer #5: Yes

5. Is the manuscript presented in an intelligible fashion and written in standard English?

Reviewer #2: Yes

Reviewer #3: (No Response)

Reviewer #4: Yes

Reviewer #5: Yes

Reviewer #2: The authors have made efforts to address earlier critiques, and the revision offers greater clarity on several points. However, I remain concerned about the level of novelty and broader impact of the study, particularly in relation to the standards of a high-impact venue such as PLOS Digital Health. In my view, the manuscript does not align with one of the journal’s core criteria: "High importance and broad interest to the community of researchers, engineers, and clinicians working in the field of digital health".

While the local cultural tailoring aspect does offer some justification, the global and regional literature increasingly shows SMS-based diabetes interventions to be feasible in the Middle East. Studies from Oman, Egypt, and earlier Saudi trials already indicate strong patient acceptance of SMS. Thus, the novelty here is relatively limited, and the study’s scope is relatively modest, and many of the feasibility findings are in line with prior literature. I remain unconvinced that this small single-arm feasibility work in 2025 is a substantial advance; the prior evidence strongly suggested feasibility. Nonetheless, the localized data may still be of value to regional stakeholders; the authors should be more explicit about how these findings materially deviate from what was "already known". Thus, while the work is methodically competent, the impact may be somewhat limited. Given the limited novelty relative to prior mHealth feasibility studies for diabetes in the region I am not fully convinced this article aligns with the mid- to high-impact nature of the journal.

Reviewer #3: Thank you again for the opportunity to review this revised manuscript. I would like to commend the authors for their thoughtful and comprehensive revisions. Many of the previously raised points have been addressed thoroughly, and the manuscript has improved significantly in clarity, structure, and adherence to PLOS Digital Health guidelines.

That said, there are still a few points I believe warrant further consideration, particularly in how some of the quantitative results are interpreted and presented:

Effect sizes and confidence intervals: While effect sizes are now included in the tables, the text often refers to “large effect size” without providing the actual value or confidence intervals. As per PLOS Digital Health’s statistical reporting standards, it would be helpful for transparency and interpretability to include the numeric effect size alongside statements of magnitude, ideally with error values if possible. This is particularly useful in feasibility studies, where significance testing is not the primary focus.

Interpretation of sitting time results: In the results section, sitting time is described as having decreased based on cumulative minutes. However, the median is unchanged, and the test result is not significant. Referencing cumulative values in this context could be misinterpreted as implying a meaningful reduction. I recommend adjusting the language to reflect the non-significance more clearly, in both the results and the discussion sections.

Narrative around non-significant results: Similarly, in the discussion, some changes (e.g., MET walking, sitting) are described directionally despite being non-significant. While it's understandable to explore trends in feasibility studies, it might be more appropriate to describe such findings as exploratory and not statistically conclusive, to avoid overinterpretation.

Figures 2 and 3 – Data Visualization: The rationale for using cumulative values in Figures 2 and 3 is noted, but presenting medians or means with variability (e.g., IQR, SD) might provide a clearer picture of the data distribution, especially given the small sample and feasibility focus. This would better align with common practice and enhance the clarity of the findings. I also still think that this should be one figure only. Repeating the same value in different units is overlapping and could be misleading.

Reviewer #4: This is a well-designed and timely feasibility study that addresses an important gap in digital health interventions for type 2 diabetes, particularly in culturally specific settings like Saudi Arabia. The research is methodologically sound, ethically conducted, and clearly written. The conclusions are appropriate and supported by the data.

The study employs the right methods for a feasibility study—primarily descriptive statistics and non-parametric tests to explore change over time. Effect sizes are reported, missing data are transparently acknowledged, and no overinterpretation of statistical results is made. The analysis is handled with care and rigor.

The data are available via a public repository, with a DOI included. This meets the journal’s standards for transparency and reproducibility.

The manuscript is clearly written, well-organized, and easy to follow. The tone is professional yet accessible, and the work is presented in a way that will resonate with both academic and clinical audiences.

I really enjoyed reading this paper. It’s a thoughtfully executed feasibility study with strong real-world relevance. The team’s commitment to improving diabetes self-management in underserved populations is evident throughout, and great care has been taken in developing and evaluating the ActiveText@T2D intervention.

Your discussion of the study limitations is particularly excellent—transparent, reflective, and forward-looking. You’ve addressed key issues like sample representativeness, potential selection bias, the evolving digital health landscape, and the limitations of SMS-based interventions with clarity and depth. These sections demonstrate a deep understanding of the complexities of conducting applied health research in real clinical settings.

The integration of patient and public involvement (PPIE) is another strength of this work, and the inclusion of both patient and nurse perspectives in your qualitative evaluation adds richness to the findings.

Overall, this is a solid and impactful contribution to digital health literature. I have no further comments or suggestions. The manuscript is ready for publication as is.

Reviewer #5: Overall Overview.

This manuscript led by Holly Blake at Nottingham University presents the feasibility and acceptability of a digital health intervention to promote physical activity in T2DM in Saudi Arabia using SMS.

Strengths of the Manuscript.

The methodology of this study follows a systematic approach based on behavioral change theories. Also, 84% recruitment and 75% retention may indicate participant’s engagement, a strong indicator for feasibility. Finaly, combining quantitative and qualitative approaches enhances the depth of the findings

Areas for Improvement.

•Though this study recruited 52 participants as sample which is good for feasibility studies, limited generalizability should be acknowledged.

•In this era, technologic is advancing fast, the authors are advised to discuss on how this digital health intervention may impact feasibility today as the intervention happened 5years ago. Highlighting that would better enhance the paper.

•To clarify on why certain messages were more impactful could be better to refine future interventions.

**Do you want your identity to be public for this peer review?** For information about this choice, including consent withdrawal, please see our Privacy Policy

Reviewer #2: No

Reviewer #3: No

Reviewer #4: **Yes: ** Nour Kassem

Reviewer #5: No

**Figure resubmission:****Reproducibility:** To enhance the reproducibility of your results, we recommend that authors of applicable studies deposit laboratory protocols in protocols.io, where a protocol can be assigned its own identifier (DOI) such that it can be cited independently in the future. Additionally, PLOS ONE offers an option to publish peer-reviewed clinical study protocols. Read more information on sharing protocols at https://plos.org/protocols?utm_medium=editorial-email&utm_source=authorletters&utm_campaign=protocols

---

## [Decision Letter · Decision Letter 2]

Response to Reviewers Revised Manuscript with Track Changes Manuscript
**Reviewers' Comments:**

**Comments to the Author**

Reviewer #2: All comments have been addressed

Reviewer #3: (No Response)

Reviewer #4: All comments have been addressed

publication criteria?

Reviewer #2: Partly

Reviewer #3: Yes

Reviewer #4: Yes

3. Has the statistical analysis been performed appropriately and rigorously?

Reviewer #2: Yes

Reviewer #3: Yes

Reviewer #4: Yes

4. Have the authors made all data underlying the findings in their manuscript fully available (please refer to the Data Availability Statement at the start of the manuscript PDF file)?

Reviewer #2: Yes

Reviewer #3: Yes

Reviewer #4: Yes

5. Is the manuscript presented in an intelligible fashion and written in standard English?

Reviewer #2: Yes

Reviewer #3: Yes

Reviewer #4: Yes

Reviewer #2: Thank you. No more comments.

Reviewer #3: Thank you once again for the opportunity to review this manuscript. I would like to sincerely commend the authors for their thoughtful and thorough revisions. The authors have significantly improved the manuscript and have addressed most of the earlier concerns. The inclusion of effect sizes and confidence intervals, as well as the more cautious tone in the discussion section, strengthen the paper and bring it closer to publication standards.

I have just one remaining concern: the use of cumulative values in the results narrative (lines 404 and 406) and in Figure 2. As currently presented, cumulative total minutes may give a misleading impression of group-level change, since they do not reflect individual-level variability or central tendencies (e.g., medians or means).

Importantly, since the authors state in the methods section that they used Wilcoxon signed-rank tests to assess pre-post differences (lines 343–344), it would be more appropriate to report medians (rather than cumulative totals) in both the results section and figures. This would ensure consistency with the statistical approach and improve interpretability.

I recommend replacing or supplementing the cumulative presentation with standard descriptive plots (e.g., boxplots or bar charts with medians and IQRs), and updating the textual descriptions to reflect medians rather than cumulative values.

This is a relatively minor revision, but an important one in terms of data clarity and methodological alignment. Once addressed, I believe the manuscript will be ready for acceptance.

Reviewer #4: Thank you for the opportunity to review the revised version of this manuscript. Although I did not submit detailed comments in the initial round, I had previously reviewed and recommended acceptance. I’m pleased to now provide written feedback, and I commend the authors for their thoughtful and thorough revisions.

This research is well grounded in behavioural theory, culturally adapted, and contextually relevant in a setting where digital health interventions remain underutilized. The revisions have significantly strengthened the manuscript, and the discussion now clearly distinguishes this study from similar efforts by emphasizing its feasibility focus, theoretical grounding, and cultural tailoring.

Strengths of the revised manuscript:

1. The use of mixed methods, including nurse and patient perspectives, strengthens the depth of findings.

2. Statistical analysis now includes effect sizes with 95% confidence intervals.

3. The limitations section is candid and comprehensive, addressing sample characteristics, timing, and generalizability.

4. Tables and figures have been improved, and the narrative around feasibility outcomes is careful and balanced.

Minor suggestions:

1. Clarify timepoints in table 4: Please add a footnote or legend that defines “T0 = baseline; T1 = 3-month follow-up”. This will assist readers in interpreting the table without needing to search earlier sections.

2. Conclude with a brief real-world summary: While the Future Directions section is already strong, I recommend adding a brief 1–2 sentence paragraph at the end of the discussion or conclusion that ties this feasibility work into the real-world healthcare infrastructure. For example, how the intervention might be integrated into national diabetes programs, primary care workflows, or broader digital health strategies in Saudi Arabia/Gulf. This would help highlight the translational value of the study beyond academic settings.

**Do you want your identity to be public for this peer review?** For information about this choice, including consent withdrawal, please see our Privacy Policy

Reviewer #2: No

Reviewer #3: **Yes: ** Shlomo Yeshurun

Reviewer #4: **Yes: ** Nour Kassem

**Figure resubmission:****Reproducibility:** To enhance the reproducibility of your results, we recommend that authors of applicable studies deposit laboratory protocols in protocols.io, where a protocol can be assigned its own identifier (DOI) such that it can be cited independently in the future. Additionally, PLOS ONE offers an option to publish peer-reviewed clinical study protocols. Read more information on sharing protocols at https://plos.org/protocols?utm_medium=editorial-email&utm_source=authorletters&utm_campaign=protocols

---

## [Decision Letter · Decision Letter 3]

The ActiveText@T2D text messaging behavioural intervention to increase physical activity in adults with type 2 diabetes: a prospective single-arm feasibility trial.

PDIG-D-25-00038R3

Dear Professor Blake,

We are pleased to inform you that your manuscript 'The ActiveText@T2D text messaging behavioural intervention to increase physical activity in adults with type 2 diabetes: a prospective single-arm feasibility trial.' has been provisionally accepted for publication in PLOS Digital Health.

Best regards,

Haleh Ayatollahi

Section Editor

PLOS Digital Health

**Additional Editor Comments (if provided):**

**Reviewer Comments (if any, and for reference):**

Reviewer's Responses to Questions

**Comments to the Author**

Reviewer #2: All comments have been addressed

Reviewer #3: All comments have been addressed

publication criteria?

Reviewer #2: Partly

Reviewer #3: Yes

3. Has the statistical analysis been performed appropriately and rigorously?

Reviewer #2: Yes

Reviewer #3: Yes

4. Have the authors made all data underlying the findings in their manuscript fully available (please refer to the Data Availability Statement at the start of the manuscript PDF file)?

Reviewer #2: Yes

Reviewer #3: Yes

5. Is the manuscript presented in an intelligible fashion and written in standard English?

Reviewer #2: Yes

Reviewer #3: Yes

Reviewer #2: No more comments.

Reviewer #3: Thank you once again for the opportunity to review this manuscript. I would like to sincerely thank the authors for their thoughtful and comprehensive responses throughout the review process. The latest revision has addressed all outstanding concerns, including the removal of the cumulative figure. This is a timely and well-executed study that makes a valuable contribution to digital health and type 2 diabetes management. I am pleased to recommend the manuscript for acceptance.

**Do you want your identity to be public for this peer review?** For information about this choice, including consent withdrawal, please see our Privacy Policy

Reviewer #2: No

Reviewer #3: **Yes: ** Shlomo Yeshurun
